# Effects of Inclusion of *N*-Carbamylglutamate in the Non-Protein Diet on Growth and Slaughter Performance, Meat Quality, Nitrogen Metabolism and Antioxidant of Holstein Bulls

**DOI:** 10.3390/ani12010033

**Published:** 2021-12-24

**Authors:** Quanyu Zhang, Guangning Zhang, Xinyue Zhang, Jinshan Yang, Yonggen Zhang

**Affiliations:** College of Animal Science and Technology, Northeast Agricultural University, Harbin 150030, China; yugezuishuai1314@yeah.net (Q.Z.); zhangguangning@neau.edu.cn (G.Z.); 18846910512@163.com (X.Z.); yjs853396960@sina.com (J.Y.)

**Keywords:** Holstein bulls, *N*-carbamylglutamate, meat quality, nitrogen metabolism

## Abstract

**Simple Summary:**

Using dietary non-protein nitrogen is an effective way to decrease the dependence on protein resources in cattle production. *N*-carbamylglutamate (NCG) is a structural analogue of *N*-acetylglutamate (NAG), which is a precursor of endogenous Arg synthesis. NCG improves urea cycling and enhances the endogenous synthesis of Arg, nitric oxide synthase and NO. The present study showed that beef benefited from being fed NCG product in the urea diet by enhancing its growth and slaughter performance, meat quality, nitrogen metabolism and plasma amino acids.

**Abstract:**

The objectives of this experiment were to investigate the effects of *N*-carbamylglutamate (NCG) on growth and slaughter performance, meat quality, nitrogen utilization, plasma antioxidant and amino acids of Holstein bulls. In this case, 24 Holstein bulls (490 ± 29.0 kg of body weights and 540 ± 6.1 d of age) were blocked by body weights and age and randomly assigned to 1 of 4 groups: (1) CON group: bulls were fed the control diet, (2) CON + NCG group: bulls were fed the control diet with 40 mg/kg BW NCG, (3) Urea group: bulls were fed the urea diet, and (4) Urea + NCG group: bulls were fed the urea diet with 40 mg/kg BW NCG. Feeding NCG significantly improved ADG, FCR, DM and CP digestibility, carcass weight, slaughter weight, DOP, eye muscle area, shear force (*p* = 0.001) and reduced L* of color, drip loss and cooking loss. Concurrently, feeding the urea diet induced a decreased ADG, carcass weight and slaughter weight, DOP, eye muscle area and shear force. NCG decreased contents of fecal N and urinary N, plasma urea in bulls and ammonia but increased N retention and utilization, plasma NO, plasma Arg, Leu, Ile and Tyr. On the other hand, feeding the urea diet increased urinary N, plasma urea and ammonia. Thus the study efficiently demonstrates that beef benefited from being fed a NCG product in the urea diet by enhancing its growth and slaughter performance, meat quality, nitrogen metabolism and plasma amino acids.

## 1. Introduction

Due to the economic and environmental considerations, the improvement of N utilization and production performance of cattle are of interest. The shortage of protein feed resources and excessive nitrogen excretion restricts the development of animal husbandry around the world. Using dietary non-protein nitrogen is an effective way to decrease the dependence on protein resources in cattle production. Ruminants can convert non-protein nitrogen to digestible microbial protein by rumen microbes. However, little information is available on the impact of non-protein nitrogen on slaughter performance, meat quality and nitrogen metabolism in Holstein bulls.

Arginine (Arg) is a functional essential amino acid and participates in the synthesis of urea, polyamines and nitric oxide (NO) [1]. Furthermore, dietary Arg enhanced bowel mucosal function in newborn and weaned pigs and improved antioxidant and immune function in rats [2,3]. However, rumen-protected and exogenous arginine impaired the absorption of alkaline amino acids and appeared to be uneconomical [4]. Thus, promoting endogenous synthesis of Arg to enhance the supply of Arg is an effective and economical strategy. *N*-carbamylglutamate (NCG) is a structural analogue of *N*-acetylglutamate (NAG), which is a precursor of endogenous Arg synthesis [5]. NCG improves urea cycling and enhances the endogenous synthesis of Arg, nitric oxide synthase and NO [6]. Recent findings demonstrated improvements in protein synthesis in skeletal muscle and mTOR signaling activity in pigs supplemented with NCG [5,7]. The increases in the milk protein yield may be related to a balanced amino acids profile, plasma NO and ammonia by promoting ureagenesis [6]. Intrauterine growth restriction in suckling lambs could be mitigated by supplying Arg and NCG through a nitric oxide-dependent pathway [8]. Dietary NCG was proved to promote energy metabolism for the damaged intestine by increasing ATP levels in the intestine of rats [9].

However, the impacts of supplementation of NCG in the urea diet on meat quality and nitrogen metabolism of Holstein bulls have rarely been studied. Although urea diets can reduce costs, low nitrogen utilization leads to environmental pollution. Therefore, the objectives of the present experiment were to explore the effects of inclusion of NCG in the basal or urea diet on growth performance of the fatting period and slaughter performance (slaughter weight, carcass weight, DOP), meat quality, nitrogen utilization, plasma antioxidant and amino acids of bulls.

## 2. Materials and Methods

### 2.1. Animals, Experimental Design and Diets

This study was performed at the Acheng Test Base of Northeast Agricultural University (Harbin, China) from April 2019 to June 2019. The experimental animal procedures recommended by the Animal Care and Use Committee (Protocol number: NEAU-[2011]-9) were approved by Northeast Agricultural University. In this case, 24 Holstein bulls (body weights; BW, 490 ± 29.0 kg and ages 540 ± 6.1 d) were used to conduct a completely randomized block design. After 2 weeks of adaptation, bulls were randomly assigned to 1 of 4 treatments according to BW and age: (1) CON group: bulls were fed the control diet, (2) CON + NCG group: bulls were fed the control diet with 40 mg/kg BW NCG, (3) Urea group: bulls were fed the urea diet, and (4) Urea + NCG group: bulls were fed the urea diet with 40 mg/kg BW NCG. Bulls were housed in individual pens (2.5 × 3 m^2^) on floor bedding with a front metal gate allowing access to feed and water.

The isoenergetic diets were formulated to nutrient requirements of bulls according to recommendations of the Beef Cattle Nutrient Requirements Model (2016), and ingredients and nutrient compositions are shown in Table 1. The NCG used in the current study (purity ≥ 97%) was obtained from ANIMORE SCI and TECH (Beijing) Co., Ltd. (Beijing, China). A top-dress supplement for each treatment was prepared by mixing 40 mg/kg BW NCG with 500 g diets, the top-dress was not mixed into the ration and was consumed readily by all cows, and the remanent was fed subsequently at 6 a.m. every day. Bulls were fed twice per day at 6 a.m. and 6 p.m. during a 7 week period. The orts of each bull were collected every morning. The offered feed was adjusted for at least 5% refusal daily according to intake of the day before.

### 2.2. Sample Collection

Body weights were recorded on the 0, 4 and 7 weeks of the experiment. Dry matter intake (DMI) was recorded daily by weighting feed offered and refused for individual bulls. Growth performance was evaluated by calculating the average daily gain (ADG) and feed conversion ratio (FCR). Samples of feed, orts, and diets were taken weekly and stored at −20 °C. Approximately 500 g of fecal samples per bull were taken from the rectum at 6 a.m. and 6 p.m. on the last three days of 0, 4 and 7 weeks. All the samples were dried at 55 °C for 48 h, ground to pass through a 1-mm screen in a Wiley mill. Urine was sampled by stimulation at 6 a.m. and 6 p.m. on the last three days of weeks 0, 4 and 7, and then acidified by H_2_SO_4_ and stored at −20 °C until analysis. Urine volume was estimated by measuring creatinine as a marker [10,11]. In calculating urine volume, we assumed creatinine output averages 29 mg/kg of BW as estimated [12]. Creatinine was detected by a colorimetric picric acid method [13]. Blood was collected from coccygeal vein by sodium heparin tubes 3 h after the morning feeding at 0, 4 and 7 weeks, and then centrifuged at 2000× *g* for 15 min at 4 °C to obtain plasma and stored at −20 °C until analysis.

### 2.3. Analysis of Nutrient Digestibility and Nitrogen Metabolism

The samples of feed and feces were examined for nutritional composition. The dry matter (DM, method 934.01), crude protein (CP, method 954.01), and ether extract (EE, method 920.39) were analyzed following the methods of AOAC International (2000) [14]. The neutral detergent fiber (NDF) was measured using a heat stable α-amylase [15]. The total tract apparent nutrient digestibility was estimated by quantifying the concentrations of indigestible NDF (iNDF) as an internal marker in the feed and feces [16]. The iNDF in the feces, feed and orts were determined by a 288 h in situ incubation [17]. Fecal N (g/d) = [CP intake (g/d) − CP intake (g/d) × % of CP Digestibility]/(% of CP in feces)/6.25; urine N (g/d) = urine volume (L/d) × [CP in urine (g/L)]/6.25 12. Nitrogen retention was calculated as N retention (g/d) = N intake − fecal N − urine N. Nitrogen utilization = (N retention/N intake) × 100.

### 2.4. Analysis of Plasma Antioxidant Capacity and Amino Acids

Bovine plasma urea and ammonia were analyzed using two-point dynamic method by a fully automatic biochemical analyzer (HT82-BTS-330, Xihuayi Technology Co., Ltd., Beijing, China). Plasma antioxidant capacity was measured with commercial kits (Nanjing Jiancheng Institute of Bioengineering, Nanjing, China). Plasma nitric oxide (NO), total antioxidant capacity (T-AOC), catalase (CAT), total superoxide dismutase (T-SOD), glutathione (GSH) were analyzed using the nitrate reductase, 2, 2′-azino-bis(3-ethylbenzothiazoline-6-sulfonic acid, visible light, hydroxylamine, thiobar-bituric and colorimetric methods following the kit structures. Plasma amino acids were analyzed by HPLC-coupled with MS detection (HPLC-LCMS/MS API3200 Q-TRAP, Thermo Fisher Scientific, Waltham, MA, USA).

### 2.5. Slaughter Performance and Meat Quality

Live weights just before slaughter were determined. At slaughter, they were stunned by captive bolt, exsanguinated and dressed. Carcass weights were determined after splitting of the carcass. The carcasses were chilled for 24 h in a room with temperature 0 °C. Dressing out percentage (DOP) was measured by the cold carcass weight divided by the live weight and then multiplied by 100. The weight of the liver, spleen and kidney were used to calculate the organ index. Meat pH and color value were measured at three spots on the longissimus dorsi. The pH value was measured using a pH meter (PHS-3C; Nanjing Nanda Analytical Instrument Application Research Institute, Nanjing, China). The meat color (L*, brightness; a*, redness and b*, yellowness) was determined using a Minolta device. The eye muscle (cm^2^) was calculated according to the cross-sectional area of longissimus dorsi between the 12th and 13th ribs. The meat was pre-weighed into a polystyrene tray with a driloc pad, over-wrapped with oxygen permeable film, and stored for 5 d at 1 °C for determination of drip loss. Cooking loss was carried out by cooking the slices in a water bath at 75 °C for 50 min. The 12.7 mm cores were sampled from the meat and sheared transversely with a Warner–Bratzler shear blade fitted to a Model 6021 Instron Universal Testing Instrument (Instron, High Wycombe, Bucking-hamshire, UK).

### 2.6. Statistical Analysis

Normality of the data was examined for normal distribution using Minitab [17] (Minitab, Inc., State College, PA, USA, 2014). All data were analyzed according to the Mixed procedure of SAS 9.4 (SAS Institute, Cary, NC, USA), included the diet (CON vs. Urea), NCG supplementation (0 vs. 40 mg/kg), diet × NCG and time were considered as fixed effects and the random effect of bull. Initial variables were used as covariates for analyses. Data were reported as least squares means and statistical significance was considered at *p* ≤ 0.05, and a tendency was considered at 0.05 < *p* ≤ 0.10. When an interaction between diet and NCG was significant, differences among the factors were analyzed by the SLICE option of LSMEANS statement.

## 3. Results

### 3.1. Growth Performance and Digestibility of Nutrients

The lower ADG (*p* < 0.01) in bulls supplemented with urea diet was observed compared to those supplemented with control diet, but supplementing NCG increased ADG of bulls fed the urea diet (*p* < 0.01). In addition, feeding NCG significantly improved ADG (*p* < 0.01) and FCR (*p* = 0.02) compared to without feeding NCG (Table 2). Digestibility of DM (*p* = 0.01) and CP (*p* = 0.02) were significantly higher in bulls fed NCG than those fed without NCG.

### 3.2. N Metabolism

Supplementing bulls with NCG significantly decreased fecal N (*p* = 0.02; Table 3). The urinary N significantly increased (*p* = 0.02) in bulls fed the urea diet, nevertheless, feeding NCG significantly decreased the urinary N (*p* = 0.02) in bulls. The N retention (*p* = 0.01) and N utilization (*p* = 0.003) significantly increased in bulls fed NCG.

### 3.3. Plasma Biochemistry and Antioxidants Index

The plasma urea significantly increased in bulls fed the urea diet (*p* = 0.02; Table 4), however, feeding NCG tended to decrease plasma urea in bulls (*p* = 0.09). Plasma ammonia increased (*p* = 0.04) when supplementing the urea diet, but the opposite response was observed (*p* = 0.02) in bulls fed NCG. The NO concentration in plasma tended (*p* = 0.09) to increase with feeding NCG.

### 3.4. Plasma Amino Acids

Regardless of urea supplementation in diets groups, bulls fed NCG had greater plasma Arg (*p* = 0.04), Leu (*p* = 0.01), Ile (*p* = 0.01) and Tyr (*p* = 0.01) than those without supplementing NCG (Table 5). An interaction (*p* = 0.01) between the urea diet and NCG was observed for content of plasma Phe. Supplementing NCG increased plasma Phe of bulls fed without urea, but the opposite response was observed when bulls were fed the urea diet. Supplementation of NCG tended to improve plasma Val (*p* = 0.07).

### 3.5. Slaughter Performance and Organ Index

Feeding a diet with NCG increased carcass weight (*p* = 0.02; Table 6). Bulls fed the urea diet had lower carcass weight (*p* = 0.02) and slaughter weight (*p* = 0.06) than those fed the control diet. Supplementing bulls with urea diet decreased DOP (*p* = 0.02), but feeding diet with NCG tended to enhance DOP of bulls (*p* = 0.08). No difference (*p* > 0.1) was present between treatments on organ index.

### 3.6. Meat Quality

Eye muscle area in bulls fed the urea diet was less than those fed the control diet (*p* = 0.001; Table 7). Eye muscle area significantly increased in bulls fed NCG (*p* = 0.001). Feeding NCG significantly decreased drip loss (*p* = 0.001) and cooking loss (*p* = 0.001). Feeding NCG significantly increased shear force (*p* = 0.001). The L* score tended to increase (*p* = 0.07) in bulls fed the urea diet, but the L* score significantly deceased (*p* = 0.03) in bulls fed NCG.

## 4. Discussion

### 4.1. Growth, N Metabolism, Plasma Antioxidants and Amino Acids

Consistent with previous studies, our results found that feeding the urea diet decreased ADG and DM and CP digestibility. The intake and growth performance of lambs were reduced by adding 8% urea to the diet [18]. Supplementing the urea diet instead of the protein feed to dairy cows also decreased milk yield [19]. This may be due to that urea can be rapidly hydrolyzed into ammonia-nitrogen in the rumen, however, rumen microbes have a low rate of synthesizing a microbial protein from ammonia-nitrogen, resulting in insufficient utilization of nitrogen in the urea diet. Supplementation of NCG induced changes in the nutrients digestibility. For instance, protein digestibility increased when bulls fed NCG. This could be explained by decreased fecal N. In addition, our study demonstrated that NCG enhanced ADG and subsequently improved FCR therefore, improving overall slaughter and carcass weight. Improved growth rate and muscle development by NCG treatment have been reported else in the literature [20]. In conclusion, the addition of NCG improved both the nitrogen utilization of the urea diet and consequently the ADG, while alleviating the difficult problem of protein feed shortage, which was supported by the synergistic effect between the urea diet and NCG. A previous study by our group suggested that supplementing Holstein bulls with 40 mg/kg of BW NCG was prior to increase growth performance compared with other doses [21]. Previous studies showed that dietary NCG had a beneficial effect on the feed efficiency of broilers and pigs [22,23]. NCG supplementation has a beneficial effect on nutrient digestion only if the dietary CP level is extremely lowered [24]. In addition, NCG improved the relative weight of the small intestine and intestinal morphology of pigs, which is beneficial for intestinal digestion 7. Arg has been recognized as a powerful factor stimulating insulin production [25]. Therefore, increased serum anabolic hormone level has been associated with greater nutrient partitioning toward processes involved in the synthesis of tissue proteins.

Feeding the urea diet increases ammonia concentrations and has been shown to decrease the efficiency of N utilization for milk and meat production. The addition of NCG decreased the urea N concentrations in plasma, milk, and urine, and resulted in a greater N conversion efficiency in dairy cows 6. Arginine is a crucial AA related to the transportation, storage, and excretion of N and the disposition of ammonia via the urea cycle [26]. Thus, urea synthesis through NCG in ruminant gut tissues might improve N utilization. The higher concentrations of plasma urea and ammonia by feeding the urea diet suggested lower efficiency of N utilization because of the imbalance of amino acids for protein synthesis and accretion. Moreover, decreased plasma urea by feeding NCG suggested higher efficiency of net protein synthesis by promoting urea cycling [27]. Therefore, feeding NCG would have beneficial impacts on dietary N utilization and N digestibility. Similar to our result, a previous study found a reduced serum urea [20]. Feeding NCG reduced concentrations of plasma ammonia and urea in dairy cows 6. Consistent with those studies, NCG decreased concentration of plasma ammonia, which suggested that NCG promoted ammonia reduction and Arg synthesis by accelerating urea recycling [28]. Arg is a crucial factor predominant in energy metabolism including fatty acids, glucose, and amino acids through nitric oxide production [29]. Feeding suckling lambs with Arg or NCG enhanced intestinal function by the operational antioxidant system by NO-signaling pathway 8. Arg can serve as a precursor of nitric oxide, creatine, ornithine, and polyamines. An increase in NO level affected the antioxidant capacity of the small bowel, thus modulating antioxidant defense [30]. Arg is a major component of the NO signaling pathway, the signaling molecules involved in the modulation of the intracellular redox environment, which plays an essential role in the biogenesis and biological function of mitochondria. Previous studies demonstrated that feeding NCG enhanced the antioxidant capacity of the liver, spleen and plasma in rats under oxidative stress [31,32]. The NCG increased plasma arginine concentration by activating carbamylphosphate synthase-1 and pyr-roline-5-carboxylate synthase, which were the key enzymes of endogenous arginine synthesis [20]. The ileal digestibility of most amino acids by supplementation of NCG is attributed to the increased endogenous amino acids synthesis [20]. The Leu can improve intestinal development and immune function by many physiological and metabolic functions [33]. A further experiment may be warranted to explore the impact of NCG on rumen function and intestinal health of beef cattle.

### 4.2. Slaughter Performance and Meat Quality

NCG has been reported to improve muscle protein synthesis of pigs by inducing the endogenous synthesis of the Arg and the Arg family of AA and the phosphorylation of mTOR [34]. The greater dressing percentage of carcasses from bulls fed NCG compared to no-supplementation of NCG with a similar slaughter weight indicated that NCG regulates bulls to consume more resources to support gain in carcass weight. Feeding NCG are capable of benefiting organ development of broiler chickens [22]. However, NCG supplementation did not influence the organ index in the current results. Presumably, feeding NCG might interact with different breeds, development stages and length of feeding time.

Improved N and amino acid metabolism can contribute to advancements in the efficient and economical production of high-quality beef. One plausible explanation for in-creased eye muscle area by NCG supplementation was possibly related to the higher Arg and Leu concentrations. Arg and Leu played an important role in muscle protein synthesis by inducing the phosphorylation of mTOR [34,35]. Feeding growing pigs with 0.1% NCG improved longissimus dorsi muscle by enhancing the endogenous synthesis of the Arg and the amino acids in the Arg family [20]. In addition, the greater ileal digestibility of most amino acids in a low-protein diet supplemented with NCG is attributed to the enhanced endogenous AA synthesis and muscle protein accretion in pigs [20]. Water holding capacity indicates processing and sensory qualities in meat characteristics, and economic losses have been caused by lower water holding capacity [36]. Drip loss can be a good indicator of water holding capacity. In agreement with our results, reduced drip loss was observed in broilers with an amniotic injection of NCG [37]. A previous study also found that reduction of drip loss was associated with declined MDA level in porcine longissimus dorsi by supplementation of NCG [37]. The reduction of cooking loss was contributed to protein solubility, primarily collagen [38]. The improvements in shear forces are attributed to the decreased collagen content and solubility in the muscle. Meat color acts as a vital indicator deciding meat quality and acceptability. The meat color would be darker because there is less free water to reflect light when proteins bind water more strongly [39]. In according to our results, previous study also suggested that the increase in lightness may be responsible for a decrease in drip loss, and reduced water holding capacity may lead to greater shear force [40].

## 5. Conclusions

Supplementing diets of NCG enhanced growth performance through improving ADG, FCR, DM and CP digestibility. Concurrently, adding NCG in urea diets decreased contents of fecal and urinary N and plasma ammonia, which in turn improved nitrogen metabolism. On the other hand, feeding NCG improved meat quality by increasing eye muscle area, shear force and L* of color, and decreased drip and cooking loss. Overall, beef benefited from being fed a NCG product in the control and urea diet. NCG is a promising feed additive in beef production.

## Figures and Tables

**Table 1 animals-12-00033-t001:** Ingredient and chemical composition of the basal diets (dry matter basis).

Parameter	the CON Diet %	the Urea Diet %
Ingredient composition, % of DM		
Corn grain	46	51
Soybean meal	5	-
Urea	-	1
Peanut hull	15	15
Soybean skin	10	10
Corn gluten feed	11	10
Corn germ meal	5	5
Molasses	5	5
Salt	0.8	0.8
Limestone	1	1
Magnesium oxide	0.5	0.5
Sodium bicarbonate	0.5	0.5
Mineral-vitamin premix ^1^	0.2	0.2
Chemical composition, % of DM		
OM	92.6	93.2
CP	12.8	13.5
NDF	34.4	34.3
ADF	18.2	18.2
EE	3.3	3.2
Ca	0.8	0.8
P	0.4	0.4
ME (MJ/kg)	11.4	11.5

^1^ The premix provided the following per kilogram of the diet: VA 2500 IU, VD 500 IU, VE 10 IU, Fe 10 mg, Cu 15.0 mg, Mn 20 mg, Zn 25 mg, I 0.50 mg, Co 0.10 mg. The percentage of minerals in the premix was 28.3%; the percentage of vitamins in the premix was 2.60%.

**Table 2 animals-12-00033-t002:** Effects of supplementation with *N*-carbamylglutamate (NCG) in urea diet on the growth performance in Holstein bulls.

Parameter	Treatment ^1^				SEM	*p*-Values		
	CON	CON + NCG	Urea	Urea + NCG		Basal Diet	NCG	Basal Diet × NCG
Growth Performance								
ADG, kg/d	1.48	1.74	1.32	1.71	0.11	<0.01	<0.01	<0.01
FCR	8.40	6.91	8.48	7.33	0.68	0.45	0.02	0.15
Intake, kg/d								
DM	12.4	12.0	11.7	12.5	0.83	0.43	0.36	0.17
CP	1.56	1.49	1.57	1.56	0.07	0.31	0.24	0.08
NDF	4.26	4.16	4.01	4.29	0.18	0.32	0.25	0.13
ADF	2.26	2.20	2.13	2.28	0.09	0.31	0.24	0.22
Digestibility %								
DM	66.9	68.7	66.6	68.6	0.68	0.78	0.01	0.88
CP	71.2	74.4	69.6	72.7	1.30	0.21	0.02	0.99
NDF	34.5	36.8	35.1	38.1	1.38	0.49	0.06	0.79
ADF	27.0	27.9	27.2	31.3	2.55	0.27	0.13	0.34

Note: ^1^ ADG, average daily gain; FCR, feed conversion ratio (kg of DMI/kg of ADG); DM, dry matter; CP, crude protein; NDF, neutral detergent fiber; ADF, acid detergent fiber. CON: feeding basic diet; CON + NCG: feeding basic diet supplemented with 40 mg/kg BW NCG daily; Urea: feeding with urea as nitrogen source diets; Urea + NCG: feeding with urea as nitrogen source diets supplemented with 40 mg/kg BW NCG daily.

**Table 3 animals-12-00033-t003:** Effects of supplementation with *N*-carbamylglutamate (NCG) in urea diet on N metabolism in Holstein bulls.

Parameter	Treatment ^1^				SEM	*p*-Values		
	CON	CON + NCG	Urea	Urea + NCG		Basal Diet	NCG	Basal Diet × NCG
N Metabolism, g/d								
N intake	249	238	251	250	4.86	0.15	0.24	0.29
Fecal N	82.2	69.9	80.5	69.9	4.55	0.85	0.02	0.86
Urinary N	105	95.2	121	105	5.16	0.02	0.02	0.57
N retention	62.2	73.3	50.1	75.8	7.07	0.50	0.01	0.31
N utilization	0.24	0.31	0.20	0.30	0.03	0.39	0.003	0.40

Note: ^1^ CON: feeding basic diet; CON + NCG: feeding basic diet supplemented with 40 mg/kg BW NCG daily; Urea: feeding with urea as nitrogen source diets; Urea + NCG: feeding with urea as nitrogen source diets supplemented with 40 mg/kg BW NCG daily.

**Table 4 animals-12-00033-t004:** Effects of supplementation with *N*-carbamylglutamate (NCG) in urea diet on the plasma biochemistry and antioxidants index in Holstein bulls.

Parameter	Treatment ^1^				SEM	*p*-Values		
	CON	CON + NCG	Urea	Urea + NCG		Basal Diet	NCG	Basal Diet × NCG
Urea, mmol/L	3.52	3.14	4.06	3.78	0.19	0.01	0.09	0.80
Plasma ammonia, µmol/L	55.1	45.5	65.4	53.7	4.33	0.04	0.02	0.81
NO, µmol/L	41.6	46.6	37.4	43.8	3.28	0.29	0.09	0.88
T-AOC, U/mL	8.98	9.01	8.89	9.03	0.72	0.96	0.91	0.94
CAT, U/mL	69.1	66.3	66.6	65.1	2.50	0.46	0.39	0.80
T-SOD, U/mL	74.4	76.7	78.9	73.8	3.07	0.80	0.65	0.23
MDA, nmol/mL	3.92	3.79	3.86	3.87	0.09	0.89	0.52	0.47
GSH, µmol/L	8.07	7.44	7.65	7.66	0.21	0.64	0.14	0.13

Note: ^1^ NO, nitric oxide; T-AOC, total antioxidant capacity; CAT, catalase; T-SOD, total superoxide dismutase; MDA, malondialdehyde; GSH, glutathione. CON: feeding basic diet; CON + NCG: feeding basic diet supplemented with 40 mg/kg BW NCG daily; Urea: feeding with urea as nitrogen source diets; Urea + NCG: feeding with urea as nitrogen source diets supplemented with 40 mg/kg BW NCG daily.

**Table 5 animals-12-00033-t005:** Effects of supplementation with *N*-carbamylglutamate (NCG) in urea diet on the plasma amino acids in Holstein bulls.

Parameter	Treatment ^1^				SEM	*p*-Values		
	CON	CON + NCG	Urea	Urea + NCG		Basal Diet	NCG	Basal Diet × NCG
EAA, µmol/L								
Arg	159	193	143	167	12.7	0.12	0.04	0.69
His	85.0	83.4	85.2	89.3	5.80	0.61	0.83	0.63
Leu	132	158	132	150	8.31	0.63	0.01	0.64
Ile	92.0	104	83.5	102	5.24	0.35	0.01	0.52
Lys	165	167	136	139	18.8	0.15	0.92	0.98
Met	31.9	31.0	33.3	33.3	2.20	0.40	0.84	0.84
Phe	40.0	47.9	46.5	37.7	2.71	0.51	0.88	0.01
Thr	59.2	73.0	73.1	67.4	10.3	0.69	0.69	0.36
Val	210	236	200	227	13.8	0.50	0.07	0.99
TEAA	974	1092	933	1012	69.8	0.39	0.17	0.78
NEAA, µmol/L								
Ala	255	236	237	230	9.16	0.21	0.18	0.53
Glu	114	108	116	103	7.26	0.82	0.20	0.63
Pro	82.8	81.2	76.9	81.1	4.49	0.52	0.78	0.52
Gly	362	381	377	413	17.9	0.21	0.13	0.65
Ser	95.9	94.3	91.2	95.3	5.42	0.74	0.82	0.61
Tyr	53.7	67.5	47.7	61.1	4.44	0.18	0.01	0.97
Cys	0.180	0.130	0.181	0.155	0.11	0.90	0.21	0.91
Asp	18.1	18.1	14.8	19.3	1.38	0.44	0.13	0.12
TNEAA	981	987	961	1003	29.4	0.94	0.43	0.54

Note: ^1^ EAA, essential amino acid; TEAA, total essential amino acid; NEAA, nonessential amino acid; TNEAA, total nonessential amino acid; Arg, arginine; His, Histidine; Leu, Leucine; Ile, Isoleucine; Lys, Lysine; Met, Methionine; Phe, Phenylalanine; Thr, Threonine; Val, Valine; Ala, Alanine; Glu, Glutamic acid; Pro, Proline; Gly, Glycine; Ser, Serine; Tyr, Threonine; Cys, Cysteine; Asp, Aspartic. CON: feeding basic diet; CON + NCG: feeding basic diet supplemented with 40 mg/kg BW NCG daily; Urea: feeding with urea as nitrogen source diets; Urea + NCG: feeding with urea as nitrogen source diets supplemented with 40 mg/kg BW NCG daily.

**Table 6 animals-12-00033-t006:** Effects of supplementation with *N*-carbamylglutamate (NCG) in urea diet on the slaughter performance and organ index in Holstein bulls.

Parameter	Treatment ^1^				SEM	*p*-Values		
	CON	CON + NCG	Urea	Urea + NCG		Basal Diet	NCG	Basal Diet × NCG
Slaughter weight (kg)	645	657	613	613	9.29	0.06	0.90	0.13
Carcass weight (kg)	357	381	336	347	10.8	0.05	0.02	0.19
DOP (%)	55.3	57.9	54.8	56.7	0.29	0.02	0.08	0.33
Organ index %								
Liver index %	1.42	1.45	1.47	1.39	0.07	0.73	0.44	0.66
Spleen index %	0.22	0.21	0.21	0.22	0.01	0.39	0.69	0.71
Kidney index %	0.25	0.27	0.26	0.27	0.02	0.30	0.73	0.86

Note: ^1^ CON: feeding basic diet; CON + NCG: feeding basic diet supplemented with 40 mg/kg BW NCG daily; Urea: feeding with urea as nitrogen source diets; Urea + NCG: feeding with urea as nitrogen source diets supplemented with 40 mg/kg BW NCG daily.

**Table 7 animals-12-00033-t007:** Effects of supplementation with *N*-carbamylglutamate (NCG) in urea diet on the meat quality in Holstein bulls.

Parameter	Treatment ^1^				SEM	*p*-Values		
	CON	CON + NCG	Urea	Urea + NCG		Basal Diet	NCG	Basal Diet × NCG
pH	6.98	6.81	6.88	6.88	0.05	0.70	0.11	0.11
Eye muscle area cm^2^	80.3	82.5	77.9	79.8	0.26	0.001	0.001	0.66
Drip loss %	4.75	3.52	4.47	3.09	0.27	0.23	0.001	0.79
Color								
L*	29.4	26.2	31.9	28.7	1.19	0.07	0.03	0.99
a*	13.3	11.9	12.9	13.4	0.45	0.44	0.49	0.18
b*	2.55	1.82	2.82	2.57	0.40	0.24	0.26	0.57
Cooking loss (%)	33.4	22.9	33.6	26.8	1.42	0.19	0.001	0.23
Shear force N	70.7	78.9	57.1	78.2	3.02	0.04	0.001	0.07

Note: ^1^ CON: feeding basic diet; CON + NCG: feeding basic diet supplemented with 40 mg/kg BW NCG daily; Urea: feeding with urea as nitrogen source diets; Urea + NCG: feeding with urea as nitrogen source diets supplemented with 40 mg/kg BW NCG daily.

## Data Availability

No new data were created or analyzed in this study. Data sharing is not applicable to this article.

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
