# Peer review of "Effects of Inclusion of N-Carbamylglutamate in the Non-Protein Diet on Growth and Slaughter Performance, Meat Quality, Nitrogen Metabolism and Antioxidant of Holstein Bulls"

_animals, 2021, doi:10.3390/ani12010033_

Round 1

Reviewer 1 Report

Introduction

Line 62: there is no slaughter perfomance. Do you mean the fattening period before slaughter?

Materials and Methods

Line 69. Please add a fullstop after university.

Line 75: Were the bulls housed individually?

Information on the NCG should be provided. Form and how it was incorporated in the diet, etc.

I find it a bit excessive that the amount of offerred feed was adjusted every day.

Table 1: Please change the title to "ingredient and chemical composition of the basal diets (dry matter basis).

Information on the metabolizable energy should be included. Were diets isoenergetic? Please add this information in Table 1.

Presentation of sample collection should follow the logical order i.e. from the live animal to the dead animal.

Line 106: did you calculate DOP on a hot or cold carcass weight. The correct practice is 24 h after slaughter.

Which muscles did you use to measure pH and colour?

Line 120: examined not determined

Line 124: amylase

Line 132: do not use abbreviations in the title

Tables 2, 3, etc. My suggestion is replace the word urea in p values with the word basal diet. What you are actually doing is compare the effect on the diets presented in Table 1.

All tables: Please replace the word item with the word parameter.

Results

The combined effect of the basal diet and NCG should be also reported.

Presentation of the results should also follow the logical order (raw to cooked).

Table 4: Colour data should be before the cooking loss.

Table 7: Please add a footnote for the abbreviations EAA, TEAA, NEAA.

Discussion

Lines 223-225. You cannot start your discussion with the results of another trial. You must rearrange the text.

The discussion is not organised well. The authos should divide the discussion in two major sections. The first one: the effect on animal growth and performance and the second one on meat quality.

Text in lines 230 -233 is a conclusion and not part of the discussion.

The interaction of the basal diet and NCG is not clearly discussed.

Conclusions: it is very confusing. There is no "take home" message. Please write again. You do not have to include all your results in the conclusions section.  Please write it again.

Author Response

RESPONSES TO REVIEWERS’ COMMENTS

Manuscript ID: animals-1479964

Title: Effects of inclusion of N-carbamylglutamate in the non-protein diet on growth and slaughter performance, meat quality, nitrogen metabolism, and antioxidant of Holstein bulls

We are truly grateful to you and the reviewers for the critical comments and thoughtful suggestions on our manuscript. They are really helpful and based on these comments and suggestions, we have revised the manuscript carefully. A native English speaker has improved the English of the manuscript. Revised portions have been marked in the manuscript. In the following pages are our point-by-point responses to the reviewers’ comments/suggestions. Please feel free to contact us if there is any question and we are very willing to improve our manuscript until all the reviewers are satisfied.

Best regards,

Yonggen Zhang

Address: College of Animal Sciences and Technology, Northeast Agricultural University, Harbin, 150030, P. R.

Tel: +86 0451 5519 0840

Fax: +86 0451 5519 0840

E-mail: zhangyonggen@sina.com

Reviewers' comments:

Reviewer: 1

Thank you for pointing out the problem. According to yours comments and suggestions, we have revised the manuscript carefully. In the following pages are our point-by-point responses to the reviewers’ comments/suggestions.

C: Line 62: there is no slaughter performance. Do you mean the fattening period before slaughter?

A: Thank you for pointing out the problem. According to your comments and suggestions,

We explained “growth and slaughter performance” in detail; the sentence “growth and slaughter performance” has been revised to “growth performance of the fatting period and slaughter performance (slaughter weight, carcass weight, DOP) ” in the revised manuscript. (Line 64-65)

C: Line 69. Please add a fullstop after university.

A: Thank you for pointing out the problem. According to your suggestion, the “.” has added to after university in the revised manuscript.(Line 70)

C: Line 75: Were the bulls housed individually?

A: Thank you for pointing out the problem. We are very sorry for our carelessness. The sentence “Bulls were housed in pens (2.5 × 3 m2) with a slatted floor” has been revised to “Bulls were housed in individual pens (2.5 × 3 m2) on floor bedding with a front metal gate allowing access to feed and water.” in the revised manuscript.(Line 78-80)

C: Information on the NCG should be provided. Form and how it was incorporated in the diet, etc.

A: Thank you for pointing out the problem. We are very sorry for our carelessness; we have added information on the NCG in line 78. And it was fed by way of “top-dress,” bulls were fed 500 g diet mixed with NCG to ensure complete feeding and the remanent was fed subsequently. The sentence “A top-dress supplement for each treatment was prepared by mixing 40 mg/kg BW NCG with 500g diets, the top-dress was not mixed into the ration and was consumed readily by all bulls, and the remanent was fed subsequently at 6 a.m. every day.” has been added in the revised manuscript.(Line 83-87)

C: I find it a bit excessive that the amount of offered feed was adjusted every day.

A: Thank you for pointing out the problem. Through careful and extensive reference review, I found the same method in another reference.

Reference:

Yang, J., Zheng, J., Fang, X., Jiang, X., Sun, Y. and Zhang, Y. Effects of dietary N-carbamylglutamate on growth performance, apparent digestibility, nitrogen metabolism and plasma metabolites of fattening Holstein bulls. Animals, 2021, 11, 126. https://doi.org/10.3390/ani11010126

C: Table 1: Please change the title to "ingredient and chemical composition of the basal diets (dry matter basis).

A: Thank you for pointing out the problem. According to your suggestion, the title “Composition and nutrient levels of basic dietary and urea dietary (dry basis)” has been revised to "ingredient and chemical composition of the basal diets (dry matter basis)” in the revised manuscript.

C: Information on the metabolizable energy should be included. Were diets isoenergetic? Please add this information in Table 1.

A: Thank you for pointing out the problem. Diets were isoenergetic in this experiment.(Line 81)And the information on the metabolizable energy has been added to table 1 in the revised manuscript.

C: Presentation of sample collection should follow the logical order i.e. from the live animal to the dead animal.

A: Thank you for pointing out the problem. According to your suggestion, we have adjusted the order of presentation of sample collection from the live animal to the dead animal in the revised manuscript.

C: Line 106: did you calculate DOP on a hot or cold carcass weight. The correct practice is 24 h after slaughter.

A: Thank you for pointing out the problem. We are very sorry for our carelessness. We calculated DOP on a cold weight. The sentence “ The carcasses were chilled for 24 h in a room with temperature 0℃. ” has been added in the revised manuscript.(Line 138-139)

C: Which muscles did you use to measure pH and colour?

A: Thank you for pointing out the problem. Meat pH and color value were measured on the longissimus dorsi muscle; the sentence “Meat pH and color value were measured at three spots on the longissimus dorsi muscle” has been added in the revised manuscript.(Line 142)

C: Line 120: examined not determined

A: Thank you for pointing out the problem. According to your suggestion, the “determined” has been revised to “examined” in the revised manuscript.

C: Line 124: amylase

A: Thank you for pointing out the problem. We are very sorry for our carelessness. The  “amylas” has been revised to “amylase” in the revised manuscript.

C: Line 132: do not use abbreviations in the title

A: Thank you for pointing out the problem. According to your suggestion, the “AAs” has been revised to “Amino acids” in the revised manuscript.

C: Tables 2, 3, etc. My suggestion is replace the word urea in p values with the word basal diet. What you are actually doing is compare the effect on the diets presented in Table 1.

A: Thank you for pointing out the problem. According to your suggestion, the “urea” in p values has been revised to “basal diet” in the revised manuscript.

C: All tables: Please replace the word item with the word parameter.

A: Thank you for pointing out the problem. According to your suggestion, the “item” in tables has been revised “parameter” in the revised manuscript.

C: The combined effect of the basal diet and NCG should be also reported.

A: Thank you for pointing out the problem. According to your suggestion, the “but supplementing NCG increased ADG of bulls fed the urea diet (P < 0.01).” has been added in results.(Line 166-167, 201-202)

C: Presentation of the results should also follow the logical order (raw to cooked).

A: Thank you for pointing out the problem. According to your suggestion, we have adjusted the order of 3.2-3.6 to follow the logical order (raw to cooked) in the revised manuscript.

C: Table 4: Color data should be before the cooking loss.

A: Thank you for pointing out the problem. According to your suggestion, we have put color data before the cooking loss in the revised manuscript.

C: Table 7: Please add a footnote for the abbreviations EAA, TEAA, NEAA.

A: Thank you for pointing out the problem. According to your suggestion, we have added a footnote for the abbreviations EAA, TEAA, NEAA, TNEAA under table 5 in the revised manuscript. (EAA: essential amino acid; TEAA: total essential amino acid; NEAA: nonessential amino acid; TNEAA: total nonessential amino acid.)

C: Lines 223-225. You cannot start your discussion with the results of another trial. You must rearrange the text.

A: Thank you for pointing out the problem. According to your suggestion, we have rearranged the text as follows “Consistent with previous studies; our results found that feeding the urea diet decreased ADG and DM and CP digestibility. The intake and growth performance of lambs were decreased by adding 8% urea in the diet [18]. ” in the revised manuscript.(Line 238-241)

C: The discussion is not organised well. The authos should divide the discussion in two major sections. The first one: the effect on animal growth and performance and the second one on meat quality.

A: Thank you for pointing out the problem. According to your suggestion, we have divided the discussion into two major groups (Growth, N metabolism, plasma antioxidants, amino acids and Slaughter performance, meat quality), the discussion has been rearranged in the revised manuscript.

C: Text in lines 230 -233 is a conclusion and not part of the discussion.

A: Thank you for pointing out the problem. According to your suggestion, we have deleted this part of the text and summed it up in the conclusion of the revised manuscript.

C: The interaction of the basal diet and NCG is not clearly discussed.

A: Thank you for pointing out the problem. According to your suggestion, “In conclusion, the addition of NCG improved both the nitrogen utilization of urea diet and consequently the ADG, while alleviating the difficult problem of protein feed shortage, which was supported by the synergistic effect between urea diet and NCG. ” has been added in the discussion.(Line 250-253)

C: Conclusions: it is very confusing. There is no "take home" message. Please write again. You do not have to include all your results in the conclusions section.  Please write it again.

A: Thank you for pointing out the problem. According to your suggestion, I have modified the conclusion to the “Supplementing diets of NCG enhanced growth performance through improving ADG, FCR, DM and CP digestibility. Concurrently, adding NCG in urea diets decreased contents of fecal and urinary N and plasma ammonia, which in turn improved nitrogen metabolism. On the other hand, feeding NCG improved meat quality. Overall, beef benefited from being fed an NCG product in the control and urea diet. NCG is a promising feed additive in beef production. ” in the revised manuscript.(Line 331-336)

We tried our best to improve the English writing and changed aspects of the revised manuscript. These changes will not influence the content and framework of the manuscript. We appreciate for editor’s and reviewers’ critical comments and thoughtful suggestions for our manuscript and hope that the revised manuscript will meet the standard of Animals.

Once again, thank you very much for your comments and suggestions.

Sincerely Yours,

Yonggen Zhang

Reviewer 2 Report

The manuscript revealed the effec of inclusion of N-carbamylglutamate in the NPN diet on the animal performance of Holstein bulls. It was found that beefs benefited from bejing fed a NCG production in the urea diet by enchancing its growth and slaughter performance , meat quality, nitrogen metabolism and plasma amino acids. THe findings can promote the application of NCG in ruminant feeds. The animal design were in good, the methods were clear. The results and discussion sounded good.

minor comments:

1) In the introduction, it is not clear why the author want to find the interaction of urea and NCG.

2) line 96, there are only two sampling time in each day. I suggest to use at least four times. Do you have any reference for two times ?

3) Table 5, I think the unit of N utilization is not true. Maybe the % need to be deleted.

Author Response

RESPONSES TO REVIEWERS’ COMMENTS

Manuscript ID: animals-1479964

Title: Effects of inclusion of N-carbamylglutamate in the non-protein diet on growth and slaughter performance, meat quality, nitrogen metabolism and antioxidant of Holstein bulls

We are truly grateful to you and the reviewers for the critical comments and thoughtful suggestions on our manuscript. They are really helpful and based on these comments and suggestions, we have revised the manuscript carefully. A native English speaker has improved the English of the manuscript. Revised portions have been marked in the manuscript. In the following pages are our point-by-point responses to the reviewers’ comments/suggestions. Please feel free to contact us if there is any question and we are very willing to improve our manuscript until all the reviewers are satisfied.

Best regards,

Yonggen Zhang

Address: College of Animal Sciences and Technology, Northeast Agricultural University, Harbin, 150030, P. R.

Tel: +86 0451 5519 0840

Fax: +86 0451 5519 0840

E-mail: zhangyonggen@sina.com

Reviewers' comments:

Reviewer: 2

Comment: The manuscript revealed the effect of inclusion of N-carbamylglutamate in the NPN diet on the animal performance of Holstein bulls. It was found that beefs benefited from Beijing fed a NCG production in the urea diet by enhancing its growth and slaughter performance, meat quality, nitrogen metabolism and plasma amino acids. The findings can promote the application of NCG in ruminant feeds. The animal design was in good, the methods were clear. The results and discussion sounded good.

Answer: Thank you for pointing out the problem. According to your comments and suggestions, we have revised the manuscript carefully. In the following pages are our point-by-point responses to the reviewers’ comments/suggestions.

C: In the introduction, it is not clear why the author want to find the interaction of urea and NCG.

A: Thank you for pointing out the problem. The addition of urea as a rapidly degradable form of nitrogen in feed can reduce its cost, but its N-utilization is low; improving the nitrogen utilization efficiency of the urea diet is a research hotspot. It was reported that NCG improves nitrogen utilization efficiency in pigs, but there are no reports on ruminants. According to your suggestion, the sentence “Although urea diets can reduce the cost, their low nitrogen utilization leads to environmental pollution” has been added in the revised manuscript.(Line 61-62)

C: line 96, there are only two sampling time in each day. I suggest to use at least four times. Do you have any reference for two times ?

A: Thank you for pointing out the problem. Through careful and extensive reference review, I found the same collection method in another reference.

Reference:

Yang, J., Zheng, J., Fang, X., Jiang, X., Sun, Y. and Zhang, Y. Effects of dietary N-carbamylglutamate on growth performance, apparent digestibility, nitrogen metabolism and plasma metabolites of fattening Holstein bulls. Animals, 2021, 11, 126. https://doi.org/10.3390/ani11010126

C: Table 5, I think the unit of N utilization is not true. Maybe the % need to be deleted.

A: Thank you for pointing out the problem. According to your suggestion, we have deleted the “ % ” in the revised manuscript.

We tried our best to improve the English writing and changed aspects of the revised manuscript. These changes will not influence the content and framework of the manuscript. We appreciate for editor’s and reviewers’ critical comments and thoughtful suggestions for our manuscript and hope that the revised manuscript will meet the standard of Animals.

Once again, thank you very much for your comments and suggestions.

Sincerely Yours,

Yonggen Zhang

Reviewer 3 Report

The article is very interesting and well performed, including the statistics. I have some minor suggestions highlighted in yellow in the attached file, for instance: replace "drop" by "drip" and so on.

Author Response

RESPONSES TO REVIEWERS’ COMMENTS

Manuscript ID: animals-1479964

Title: Effects of inclusion of N-carbamylglutamate in the non-protein diet on growth and slaughter performance, meat quality, nitrogen metabolism and antioxidant of Holstein bulls

We are truly grateful to you and the reviewers for the critical comments and thoughtful suggestions on our manuscript. They are really helpful and based on these comments and suggestions, we have revised the manuscript carefully. A native English speaker has improved the English of the manuscript. Revised portions have been marked in the manuscript. In the following pages are our point-by-point responses to the reviewers’ comments/suggestions. Please feel free to contact us if there is any question and we are very willing to improve our manuscript until all the reviewers are satisfied.

Best regards,

Yonggen Zhang

Address: College of Animal Sciences and Technology, Northeast Agricultural University, Harbin, 150030, P. R.

Tel: +86 0451 5519 0840

Fax: +86 0451 5519 0840

E-mail: zhangyonggen@sina.com

Reviewers' comments:

Reviewer: 3

Comment: The article is very interesting and well performed, including the statistics. I have some minor suggestions highlighted in yellow in the attached file, for instance: replace "drop" by "drip" and so on.

Answer: According to your comments and suggestions, we have revised the manuscript carefully. In the following pages are our point-by-point responses to the reviewers’ comments/suggestions.

C: L24- drop loss

A: Thank you for pointing out the problem. We are very sorry for our carelessness. The “drop loss” has been revised to “drip loss” in the revised manuscript. (Line 25)

C: L42- ni-trogen

A: Thank you for pointing out the problem. We are very sorry for our carelessness. The “ni-trogen” has been revised to “nitrogen” in the revised manuscript. (Line 43)

C: L63- antioxi-dant

A: Thank you for pointing out the problem. We are very sorry for our carelessness. The “antioxi-dant” has been revised to “antioxidant” in the revised manuscript. (Line 66)

C: L69- Twenty-four

A: Thank you for pointing out the problem. We are very sorry for our carelessness. The “Twenty-four” has been revised to “Twenty four” in the revised manuscript. (Line 72)

C: L75- 2.5×3 m2

A: Thank you for pointing out the problem. We are very sorry for our carelessness. The “2.5×3 m2” has been revised to “2.5×3 m2” in the revised manuscript. (Line 79)

C: L79-L80 L93-L94 L96- 0600 h, 1800 h

A: Thank you for pointing out the problem. We are very sorry for our carelessness. The “0600 h and 1800 h” has been revised to “6 a.m. and 6 p.m.” in the revised manuscript. (Line 87-88,102,104)

C: L90- recoded

A: Thank you for pointing out the problem. We are very sorry for our carelessness. The “recoded” has been revised to “recorded” in the revised manuscript. (Line 98)

C: L97- H2SO4

A: Thank you for pointing out the problem. We are very sorry for our carelessness. The “H2SO4” has been revised to “H2SO4” in the revised manuscript. (Line 105)

C: L100- were

A: Thank you for pointing out the problem. We are very sorry for our carelessness. The “were” has been revised to “was” in the revised manuscript. (Line 108)

C: L112- deter-mined and cm2

A: Thank you for pointing out the problem. We are very sorry for our carelessness. The “deter-mined and cm2” has been revised to “determined and cm2” in the revised manuscript. (Line 145)

C: L115- perme-able

A: Thank you for pointing out the problem. We are very sorry for our carelessness. The “perme-able” has been revised to “permeable” in the revised manuscript. (Line 148)

C: L124- α-amylas

A: Thank you for pointing out the problem. We are very sorry for our carelessness. The “α-amylas” has been revised to “α-amylase” in the revised manuscript. (Line 116)

C: L180- in-crease

A: Thank you for pointing out the problem. We are very sorry for our carelessness. The “in-crease” has been revised to “increase” in the revised manuscript. (Line 228)

C: L193- N metab-olism

A: Thank you for pointing out the problem. We are very sorry for our carelessness. The “N metab-olism” has been revised to “N metabolism” in the revised manuscript. (Table 3)

C: L266- Drop loss

A: Thank you for pointing out the problem. We are very sorry for our carelessness. The “Drop loss” has been revised to “Drip loss” in the revised manuscript. (Line 317)

C: L269- driping loss

A: Thank you for pointing out the problem. We are very sorry for our carelessness. The “driping loss” has been revised to “drip loss” in the revised manuscript. (Line 320)

C: L270- The reduced in cooking loss was contributed to protein solubility

A: Thank you for pointing out the problem. We are very sorry for our carelessness. The “The reduced in the cooking loss was contributed to protein solubility” has been revised to “The reduction of cooking loss was contributed to protein solubility” in the revised manuscript. (Line 321-322)

C: L274- In ac-cording

A: Thank you for pointing out the problem. We are very sorry for our carelessness. The “In ac-cording” has been revised to “In according” in the revised manuscript. (Line 326)

C: L313- drop

A: Thank you for pointing out the problem. We are very sorry for our carelessness. The “drop” has been revised to “drip” in the revised manuscript. (Previous conclusions have been deleted)

We tried our best to improve the English writing and changed aspects of the revised manuscript. These changes will not influence the content and framework of the manuscript. We appreciate for editor’s and reviewers’ critical comments and thoughtful suggestions for our manuscript and hope that the revised manuscript will meet the standard of Animals.

Once again, thank you very much for your comments and suggestions.

Sincerely Yours,

Yonggen Zhang

Round 2

Reviewer 1 Report

Conclusions

Please be more specific on the beneficial effects of NCG on meat quality. 

Author Response

RESPONSES TO REVIEWERS’ COMMENTS

Manuscript ID: animals-1479964

Title: Effects of inclusion of N-carbamylglutamate in the non-protein diet on growth and slaughter performance, meat quality, nitrogen metabolism and antioxidant of Holstein bulls

We are truly grateful to you and the reviewers for the critical comments and thoughtful suggestions on our manuscript. They are really helpful and based on these comments and suggestions, we have revised the manuscript carefully. A native English speaker has improved the English of the manuscript. Revised portions have been marked in yellow in the manuscript. In the following pages are our point-by-point responses to the reviewers’ comments/suggestions. Please feel free to contact us if there is any question and we are very willing to improve our manuscript until all the reviewers are satisfied.

Best regards,

Yonggen Zhang

Address: College of Animal Sciences and Technology, Northeast Agricultural University, Harbin, 150030, P. R.

Tel: +86 0451 5519 0840

Fax: +86 0451 5519 0840

E-mail: zhangyonggen@sina.com

Reviewer:

Thank you for pointing out the problem. According to your comments and suggestions, we have revised the manuscript carefully.

C: Conclusions Please be more specific on the beneficial effects of NCG on meat quality.

A: Thank you for pointing out the problem. According to your suggestion, the sentence “ On the other hand, feeding NCG improved meat quality ” has been revised to “On the other hand, feeding NCG improved meat quality by increasing eye muscle area, shear force and L* of color, and decreased drip and cooking loss.” in the revised manuscript.(Line 334-335)

We tried our best to improve the English writing and changed aspects of the revised manuscript. These changes will not influence the content and framework of the manuscript. We appreciate for editor’s and reviewers’ critical comments and thoughtful suggestions for our manuscript and hope that the revised manuscript will meet the standard of Animals.

Once again, thank you very much for your comments and suggestions.

Sincerely Yours,

Yonggen Zhang